# Terahertz Refractive Index and Temperature Dual-Parameter Sensor Based on Surface Plasmon Resonance in Two-Channel Photonic Crystal Fiber

**DOI:** 10.3390/s24196225

**Published:** 2024-09-26

**Authors:** Doudou Wang, Wenchuan Guo, Yizu Zou, Tian Ma, Weifeng Wang, Guoxiang Chen

**Affiliations:** 1College of Sciences, Xi’an University of Science and Technology, Xi’an 710054, China; wenchuan_guo@163.com (W.G.); 13849792516@163.com (Y.Z.); 2College of Safety Science and Engineering, Xi’an University of Science and Technology, Xi’an 710054, China; tianma@xust.edu.cn (T.M.); wangwf@xust.edu.cn (W.W.); 3College of Sciences, Xi’an Shiyou University, Xi’an 710065, China; guoxchen@xsyu.edu.cn

**Keywords:** terahertz, surface plasmon resonance, graphene, refractive index sensor, temperature sensor

## Abstract

A terahertz photonic crystal fiber with two sensing channels was designed. Graphene coated on the micro-grooves in the cladding was used as plasma material to introduce tunability. The dispersion relation, mode coupling, and sensing characteristics of the fiber were studied using the finite element method. Ultrahigh sensitivity of 2.014 THz/RIU and 0.734 GHz/°C were obtained for analytes with refractive index in the range of 1.33 to 1.4 and environment temperature in the range of 10–60 °C, respectively. Refractive index resolution can reach the order of 10^−5^. The dual parameter simultaneous detection, dynamic tunable characteristics, and working in the low-frequency range of terahertz enable the designed photonic crystal fiber to have application prospects in the field of biosensing.

## 1. Introduction

Terahertz (THz) radiation with frequency in the range of 0.1 THz to 10 THz has potential applications in bio/chem-sensing as the vibration and rotation modes of many biological macromolecules located in the THz region [1,2,3]. Sensing technology based on the surface plasmon resonance (SPR) effect has many superior performances, such as high-sensitivity, high-resolution, label-free, and real-time detection of biological analytes [4,5]. Photonic crystal fiber (PCF) with porous microstructured cladding can be used as a waveguide and sample cell for bio/chem-sensors, which are robust, compact, and biocompatible.

PCF-based SPR sensors have attracted a lot of interest owing to their unique advantages, such as flexible structure, ease of integration, and remote sensing capability compared with the traditional prism-based SPR sensors [6,7]. However, most of the reported PCF-based SPR sensors work in the communication or mid-IR band due to the commonly used plasmonic materials (i.e., noble metals or metal oxides possess superior performance in these frequencies) [7,8,9,10]. Compared with the fiber-based SPR sensor working the optical band, its counterpart in the THz range possesses many superiorities such as robust structure, ease of operation, the greater penetration depth of the evanescent field, sensitivity to larger particles such as bacteria, the capability of penetrating non-transparent biological samples, and improving the sensitivity of biomolecules [11,12]. So THz sensors have more advantages in the field of biosensing [13].

Although there are many studies about the THz PCF sensors, most are based on the nonresonant and direct absorption configurations [14]. The first THz PCF-based SPR sensor was reported by Hassani and Skorobogatiy [11]. Ferroelectric polyvinylidene fluoride (PVDF) layer wrapped around a porous Teflon fiber was used as the plasmonic material. A spectral sensitivity of 400 μm/RIU and resolution on the order of 10−4 (assuming 100 MHz spectral resolution of a THz time-domain spectroscopy setup) is predicted around 1 THz for gaseous analytes. Liu et al. proposed a Zeonex-based dual-channel SPR PCF sensor with a PVDF layer coated on the inner surface of the channels [15]. Spectral sensitivity of 110 μm/RIU and resolution on the order of 10^−4^ (assuming a 1% change in emission intensity can be detected) is obtained around 1 THz for gaseous analytes. D-shaped THz PCFs using MoS_2_ and PVDF layers as plasmonic materials, respectively, have been reported for external biosensing [12,13]. Liu et al. proposed a THz PCF-based biosensor [16]. Silver nanopillars were adopted to excite the SPR effect, and a high spectral sensitivity of 1.749 THz/RIU was obtained. However, dynamic tunability, which is very important for practical applications, has not been studied in these reported THz PCF-based SPR sensors. In our previous papers, tunable THz PCFs based on graphene plasmonics have been studied for refractive index (RI) sensing [17,18].

It has been reported that 2D material graphene possesses intrinsic plasmons that are tunable and adjustable [19]. Novel devices based on graphene plasmonics can work in a wide frequency range (i.e., from terahertz to the visible) and possess low power consumption and compact sizes. Theoretical research indicates that graphene is a good plasmonic material in the THz band [20]. Although graphene has been used to reduce oxidation of the metallic material, introduce tunability, and improve sensitivity [21,22], there are few reports about PCF sensors-based graphene plasmonics. Paul, A.K. et al. proposed an SPR sensor working in the visible spectrum coated with graphene on the outside of PCF and used directly as the plasmonic material [23]. However, tunable THz PCF sensors based on graphene plasmonics have not been reported by other groups to the best of our knowledge. What’s more, it is almost blank for simultaneous measuring of the RI and temperature by using a PCF-based SPR sensor in the terahertz region, although the RI and temperature are important properties of substances, and it is necessary to measure them simultaneously.

In this paper, we propose a tunable THz two-channel PCF based on graphene plasmonics (Gr-PCF) for simultaneous measuring of the RI and temperature. Sensing performances are investigated theoretically by using the finite element method (FEM) [24]. Ultrahigh RI sensitivity and high figure of merit (FOM) value are obtained by designing the two-sided selective filling PCF structure. What’s more, the proposed sensor works in the lower frequencies of the THz band where the water absorption loss is relatively low, so it is suitable for liquid biological analytes.

## 2. Fiber Structure Design and Numerical Method

A cross-section of the designed two-channel Gr-PCF is shown in Figure 1. The cladding consists of two rings of circular holes arranged in a triangular lattice (with lattice constant *Λ*) and two sensing channels (channels A and B). Two holes in the second ring were side-polished to form two micro-grooves (with radius r) for depositing or transferring the graphene layer and loading the analytes. The micro-grooves can reduce the distance between the core and the graphene and thus obtain a stronger SPR effect. The channel pillars with the thickness of *t* can suspend the fiber and isolate the sensing channels. TOPAS cyclic olefin copolymer (COC), which has low absorption loss, low material dispersion, low water absorption, and excellent heat and corrosion resistance, was adopted as the base material [25]. Graphene was coated on the surface of the micro-grooves as plasma material. To compare the coupling effect, two different filling types were designed and denoted as Type 1 and Type 2. Channel A and channel B were filled with temperature-sensitive liquid and analytes, respectively, for Type 1 as shown in Figure 1a. Two cladding holes adjacent to the micro-grooves were also filled with the same material for Type 2, as shown in Figure 1b, named the two-sided selective filling structure. The following structural parameters were used for the designed Gr-PCF: *Λ* = 300 μm, *d*_1_ = 100 μm, *d*_2_ = 150 μm, *H* = 450 μm, *r* = 300 μm, *R* = 750 μm and *t* = 50 μm.

The full-vector FEM was used to simulate the modes properties and loss properties of the designed THz Gr-PCF. A Perfectly Matched Layer (PML) is set at the boundary condition to absorb the radiant energy and make the calculation results more accurate. The monolayer graphene with a thickness of *t*_g_ = 0.34 nm can be effectively treated as a 2D surface conductivity. For the terahertz region, the surface conductivity of graphene is predominantly determined by the intra-band transition and can be approximately given by the Kubo formula [26]:(1)$$\sigma_{\mathrm{intra}}\left(\omega ,\mu_{c},\tau ,T\right)=\frac{ie^{2}k_{B}T}{\pi \hbar^{2}\left(\omega +i\tau^{-1}\right)}\left[\frac{\mu_{c}}{k_{B}T}+\right.\left.2\ln\left(\exp\left(-\frac{\mu_{c}}{k_{B}T}\right)+1\right)\right],$$
where *ω* = 2π*f* is the angular frequency of the incident terahertz wave, *μ*_c_ represents the graphene chemical potential (Fermi energy), *τ* is the carrier relaxation time, which is assumed as 3 ps, and *T* is temperature. *e*, *k*_B,_ and ħ are the electron charge, reduced Planck’s constant, and Boltzmann’s constant, respectively.

Considering *ħω* << 2*μ*_c_ and *k*_B_*T* << *μ*_c_ in our simulation, the surface conductivity of graphene can be further simplified by the Drude model and expressed as [26]:(2)$$\sigma_{\mathrm{intra}}\left(\omega ,\mu_{c},\tau \right)=\frac{ie^{2}\mu_{c}}{\pi \hbar^{2}\left(\omega +i\tau^{-1}\right)},$$

The relative complex permittivity of graphene can be expressed as a function of the surface conductivity [26]:(3)$$\varepsilon_{g}=1+\frac{i\sigma_{\mathrm{intra}}}{\omega \varepsilon_{0}t_{g}},$$
where *ε*_0_ is the permittivity of vacuum and *t*_g_ is the thickness of graphene.

The RI of the mixed temperature-sensitive liquid composed of ethanol and chloroform can be defined as [27]:(4)$$n_{m}=x\%\times \left[n_{ethanol}{|}_{T=20{}^{\circ}C}+\frac{dn_{ethanol}}{dT}\times (T-20)\right]+(100-x)\%\left[n_{chloroform}{|}_{T=20{}^{\circ}C}+\frac{dn_{chloroform}}{dT}\times (T-20)\right],$$
where x% and (100 − x)% are the percentages of ethanol and chloroform in the mixed liquid, respectively. dn/d*T* denotes the thermal-optical coefficient which is −3.94 × 10^−4^/°C and −6.328 × 10^−4^/°C for ethanol and chloroform, respectively. The RI of ethanol and chloroform are 1.36048 and 1.43136, respectively, when *T* = 20 °C [27]. The temperature-sensitive liquid is a mixture of 50% alcohol and 50% chloroform. The thermal-optical coefficient of Topas (1 × 10^−4^ RIU/°C) is also considered for more accurate simulation results [28]. Since the boiling point of ethanol and chloroform are 78.4 °C and 61.3 °C, respectively, the temperature of the mixed liquid can be as high as 60 °C.

Loss of the SPR-based PCFs near the resonant frequency and position of the loss peak is mainly determined by the confinement loss (CL) due to the coupling of the fundamental core mode to the surface plasmon polariton (SPP) mode, causing significant energy leakage. The sensing characteristics of SPR-based transmissive fiber sensors can be characterized by the spectral or intensity measurement. The frequency sensitivity based on spectral measurement can be expressed as
(5)$$S_{f}(THz/RIU)=\frac{\Delta f_{R}}{\Delta n_{a}},$$
where Δ*f*_R_ and Δ*n*_a_ denote the change in frequency and the RI, respectively. For the PCF-based SPR sensor with a length of 1/α and with respect to the small changes of *n*_a_, the amplitude sensitivity based on the intensity measurement method can be expressed as:(6)$$S_{A}(RIU^{-1})=-\frac{1}{\alpha (f,n_{a})}\frac{\partial \alpha (f,n_{a})}{\partial n_{a}},$$
where $\alpha \left(f,n_{a}\right)$ is the propagation loss, and $\partial \alpha \left(f,n_{a}\right)$ is loss difference. Replacing the *n_a_* in Equations (5) and (6) with the temperature *T* will give the frequency sensitivity and amplitude sensitivity for temperature sensing, respectively.

Resolution is an important parameter of sensors [29]. The RI resolution of a sensing system based on frequency spectrum measurement can be expressed as
(7)$$R=\Delta n_{a}\frac{\Delta f_{\min}}{\Delta f_{R}}$$
where Δ*n*_a_ denotes the change in the analyte RI, Δ*f_R_* is the corresponding shift of the resonance frequency caused by the change of the analyte RI, Δ*f*_min_ denotes the minimum spectral resolution of the detector. The RI resolution can also be expressed as R = Δ*f*_min_/*S_f_* by substituting Equation (5) into Equation (7), which depends on the sensitivity of the sensor and the frequency spectral resolution of the THz-TDS setup.

## 3. Results and Discussion

### 3.1. Dispersion Relation and Mode Coupling Characteristics

Figure 2a,b show the mode dispersion curves and loss spectra of the two-channel Gr-PCF with filling type 1 and filling type 2, respectively. When the effective refractive index Re (neff) of the SPP mode is equal to that of the core mode (i.e., their dispersion curves intersect), resonance coupling occurs between them, and the corresponding frequency is defined as the resonance frequency fR. At the resonance frequency, the energy of the core mode couples into the SPP mode and decays rapidly, which results in a significant decrease of energy in the core region. The position of the loss peaks or the resonance frequency changes with the RI of the filling material. Therefore, RI sensing can be achieved by monitoring the position and movement of the loss peaks. From this point of view, sharper loss peaks (weaker core energy) mean better sensing performance (i.e., high FOM value of the sensor). It can be seen that the loss spectra of filling type 2 (i.e., the two-sided selective filling structure) have sharper loss peaks and, thus, superior sensing performance. By comparing the mode profiles of Figure 2(i) with Figure 2(iii) or Figure 2(ii) with Figure 2(iv), it can be seen that the filling type 2 has more core mode energy coupled into the SPP mode (i.e., higher coupling efficiency and thus sharper loss peaks). Because the filling type 2 can achieve complete coupling between the core mode and SPP mode. Besides, the filling type 2 has more energy located in the cladding holes filled with temperature-sensitive liquid or analytes. The SPP modes are mainly located in the dielectrics on both sides of the plasmonic material and are extremely sensitive to the changes in the RI of these dielectrics. From this point of view, the two-sided filling structure is capable of providing better sensing performance compared with the filling type 1. So, type 2 was adopted in the subsequent study. The peak A and peak B in the loss spectra are caused by the coupling of the fundamental core mode to the SPP modes in channel A and channel B, respectively, as indicated by the mode profiles in Figure 2(iii) and Figure 2(iv). It can be seen from Figure 2(v) that the energy of the core mode is well confined in the core region at the non-resonance frequency. The position and intensity of the loss peak A at the lower frequency (0.552 THz) is sensitive to the RI as well as T of the temperature-sensitive liquid in the channel A, while loss peak B at the higher frequency (0.905 THz) is sensitive to the RI of the analyte in the channel B. Dual parametric sensing of temperature and the RI can be realized by detecting the position and intensity of the two peaks in the loss spectrum.

### 3.2. Dynamic Tunable Characteristics

For conventional PCF SPR sensors with noble metal as the plasma material, the working band is determined once it is prepared. However, the working band of Gr-PCF SPR sensors is dynamically tunable as the resonance frequency is related to the chemical potential of graphene, which can be adjusted by the bias voltage [30]. Figure 3a shows the loss spectra of the designed Gr-PCF with different *µ*_c_. It can be seen the resonance frequencies in both channels are different; even their *µ*_c_ are the same owing to the different RI in channel A (*T* = 30 °C) and Channel B (*n*_a_ = 1.33), as indicated by the green line. The *f*_R_ in both channels increases linearly with *μ*_c,_ as shown in Figure 3b. To reduce the interaction between the two loss peaks and facilitate the simultaneous dual-parameter measurement, the loss peaks can be further separated by increasing the difference of *µ*_c_ between the two channels. In addition, the working band of the designed two-channel Gr-PCF can be dynamically tuned to the frequency of interest or the lower frequency of the THz band where the water absorption loss is relatively low.

### 3.3. Two-Channel Sensing Characteristics

The sensing characteristics with the RI and temperature in the range of 1.33–1.4 and 10–60 °C, respectively, were investigated. Figure 4a shows loss spectra of the two-channel Gr-PCF with different RIs of the analyte in channel B and T of the temperature-sensitive liquid in channel A was set as 30 °C. It can be seen that the loss peak B at higher frequencies is caused by the SPR effect in channel B redshifts significantly with an increase in *n_a_*, while the position of the loss peak A at lower frequencies remains almost unchanged. The resonance frequency of channel B decreases linearly (with R^2^ = 0.998) from 0.905 THz to 0.764 THz as *n_a_* increases from 1.33 to 1.4, and a high RI sensitivity of 2.014 THz/RIU (corresponding to 873.98 μm/RIU) is obtained as shown in Figure 4b. By monitoring the position and movement of the loss peak B in the transmission spectrum, it is possible to detect the value and change of the RI of the analyte. The RI resolution of the order of 10^−5^ can be obtained by using a time domain THz setup with 100 MHz spectral resolution. In addition to sensitivity, FOM and Q-factor are also important parameters describing the performance of the sensor, which are defined as the ratio of sensitivity and center frequency to FWHM, respectively. Due to the high sensitivity, narrower spectral lines, and low operating frequency, the designed sensor has a good sensing performance with an FOM and Q-factor of 80.2 and 36.2, respectively, when *n_a_* = 1.33.

The sensing performance of the designed sensor can also be characterized by the simple and frequency-independent intensity detection method as given by the amplitude sensitivity SA. Figure 5a shows the amplitude sensitivity of channel B within different RI ranges. The maximum amplitude sensitivity of 123 RIU^−1^ was obtained at 0.886 THz when *n_a_* is in the range of 1.33–1.34. Assuming that a 1% change in the intensity of the transmitted terahertz waves can be reliably detected, an RI resolution of the order of 10^−5^ is predicted for the designed sensor. Figure 5b shows the power flow along the x-axis (the red lines across the mode field) of the fundamental core mode at the resonance frequency of 0.905 THz for *n_a_* = 1.33. It can be seen that when detected around 0.905 THz, the power flow in the core region has a minimum value when *n_a_* = 1.33 and increases with the increase of *n_a_*. When *n_a_* increased from 1.33 to 1.34, which contains most of the liquid-phase bioanalytes [12], the intensity of the fundamental mode at the core area increased by more than 80%.

The RI sensing performance of the proposed sensor is compared with the reported THz PCF-based SPR sensors, which are listed in Table 1. The proposed sensor has superior RI sensing performance, dual parameter simultaneous detection ability, and dynamic tunability compared with the ever-reported THz PCF SPR sensors. Moreover, compared with the sensors reported by other groups [11,12,13,14,15,16], the proposed sensor works in lower frequencies (less than 1 THz) where the absorption loss of water and base material is relatively low compared to frequencies higher than 1 THz.

Figure 6 shows the loss spectra of the two-channel Gr-PCF with different *T* of the temperature-sensitive liquid in channel A, and the RI of the analyte in channel B was set as 1.33. It can be seen that the loss peak A at lower frequencies is caused by the SPR effect in channel A blueshifts with an increase in *T*, while the position of the loss peak B at higher frequencies remains unchanged. The resonance frequency of channel A increases linearly (with R^2^ = 0.99865) from 0.537 THz to 0.574 THz as *T* increases from 10 °C to 60 °C, and the temperature sensitivity is 0.734 GHz/°C. A temperature resolution of 0.136 °C can be obtained by using a time domain THz setup with 100 MHz spectral resolution. Figure 7a shows the amplitude sensitivity of channel A within different temperature ranges. The maximum amplitude sensitivity of 0.0156 °C^−1^ is obtained when *T* is in the range of 50–60 °C. Figure 7b shows the power flow along the x-axis of the core mode at the resonance frequency of 0.537 THz for *T* = 10 °C. It can be seen that when detected around 0.537 THz, the power flow in the core region has a minimum value when *T* = 10 °C, and increases linearly with the increasing of *T*.

Temperature sensitivity is mainly determined by the thermal-optical coefficient of the temperature-sensitive material. The temperature-dependent optical constants of several temperature-sensitive materials such as strontium titanate (SrTiO_3_, STO) [31], polydimethylsiloxane (PDMS) [32], and liquid crystal (LC) [33] have been studied in the terahertz frequency range. The LC 5CB has a low extinction coefficient and larger temperature gradient of the ordinary refractive index dno/dT around room temperature at terahertz frequencies [33,34]. The temperature sensing characteristics of the proposed Gr-PCF with 5CB as temperature-sensitive material were studied and shown in Figure 8. The frequency dependence and the temperature dependence of the THz optical constants of nematic LC 5CB were based on the experimental results of Pan et al. [34]. The two cladding holes adjacent to the micro-grooves of channel A were filled with LC 5CB, and the RI of the analyte in channel B was set as 1.33. It can be seen that the resonance frequency of channel A decreases with the increase in temperature, while the resonance frequency of channel B remains almost unchanged. The maximum temperature sensitivity of 6.0 GHz/°C was obtained in the temperature range of 33 °C to 34 °C. The results show that the temperature sensitivity of the designed sensor can be improved by using a filling material with a higher thermal-optical coefficient. However, the mixture of ethanol and chloroform was still used as the temperature-sensitive material in the subsequent study as it has advantages such as ease of filling a linear and wider temperature response range.

Figure 9 shows loss spectra of the core mode with different temperatures and *n_a_*, i.e., the dual-parameter sensing effect of the designed fiber. Temperature and RI sensing can be realized simultaneously by frequency or intensity detection. Temperature and RI sensing were realized using the lower frequency region of 0.5–0.6 THz and the higher frequency region of 0.7–1.0 THz, respectively. The thermal optical effect of the base material has been considered during the simulation process. Therefore, the designed sensor is capable of measuring the values and changes of the environmental temperature and RI of the analyte, as well as the temperature dependence of the RI in practical applications.

Most of the reported PCF-based two-channel sensors work in the communication band, while dual-parameter sensors for the RI and temperature in the THz band are mainly based on metamaterials. The sensing performance of the proposed dual-parameter sensor is compared with the reported terahertz RI and temperature sensors, which are listed in Table 2. It can be seen that the proposed sensor has superior RI sensing performance (i.e., ultrahigh RI sensitivity and relatively high FOM and Q-factor), although its temperature sensing sensitivity is relatively low, which can be improved by adopting temperature-sensitive materials with high thermal-optical coefficients. Moreover, the proposed sensor works in the lower frequency band (less than 1 THz) with low water absorption loss, which is suitable for biosensing applications.

### 3.4. Preparation Method Discussion

To evaluate the fault tolerance during the preparation process of the designed two-channel Gr-PCF, the cross-section of the fiber was scaled by ±0.5% and ±1%, and the corresponding loss spectra of the core mode were shown in Figure 10. When the cross-section was changed by ±1%, the variations of the hole diameter *d*_1_ and *d*_2_ were 1 μm and 1.5 μm, respectively. It can be seen that the resonance frequencies and loss peaks change very little. In addition, misalignment of hole position has little impact on the transmission performance owing to the refractive index guiding mechanism of the designed fiber.

It is possible to fabricate the designed THz two-channel PCF directly by the existing techniques, such as extrusion, polymer jetting [39], or 3D printing prototyping [40]. Although it is challenging work to prepare a high-quality graphene layer in the channels of polymer PCF, transferring the graphene film made by the CVD method to the inner wall of the PCF channels or by the oxidation-reduction method has been reported [41]. Besides, the proposed THz two-channel sensor can also be prepared by encapsulation and isolation of the open channel Gr-PCF, as shown in Figure 11.

## 4. Conclusions

In this paper, a two-channel THz sensor based on PCF and graphene SPR is designed to detect refractive index and temperature simultaneously. Owing to the use of graphene as the plasma material, the operating frequency of this sensor has dynamic tunable characteristics. The results show a high-frequency sensitivity of 2.014 THz/RIU in the refractive index range of 1.33–1.4 using spectral detection. Using the intensity detection method, the maximum amplitude sensitivity was 123 RIU^−1^ (at 0.886 THz) in the refractive index range of 1.33–1.34. Both detection methods for refractive index sensing have resolutions up to 10^−5^ orders of magnitude. It has a temperature sensitivity of 0.734 GHz/°C over the temperature range of 10–60 °C and a temperature resolution of 0.136 °C using a terahertz time-domain spectral system with a spectral resolution of 100 MHz. The designed two-channel sensor operates in the frequency band of less than 1 THz with low water absorption loss, which is advantageous in biosensing.

## Figures and Tables

**Figure 1 sensors-24-06225-f001:**
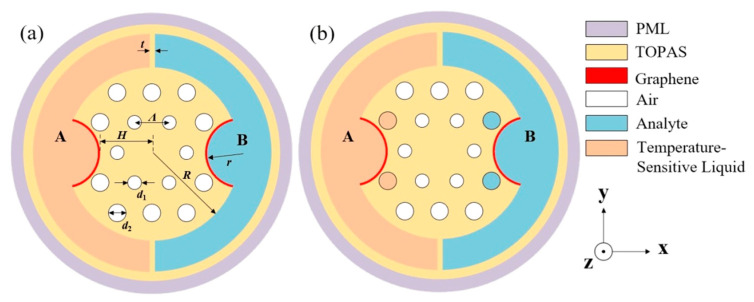
Cross-section of the designed two-channel Gr-PCF (**a**) Type 1, (**b**) Type 2.

**Figure 2 sensors-24-06225-f002:**
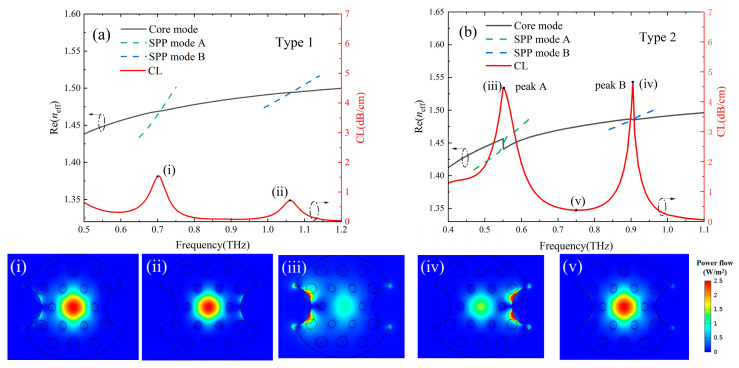
Mode dispersion curves and loss spectra of the two-channel Gr-PCF (**a**) Type 1, (**b**) Type 2. The lower panels show mode profiles of the core modes corresponding to the frequencies marked with the small black dots on the loss spectra (Channel A: *T* = 30 °C, *μ*_c_ (A) = 0.6 eV, Channel B: *μ*_c_ (B) = 1.1 eV, *n_a_* = 1.33).

**Figure 3 sensors-24-06225-f003:**
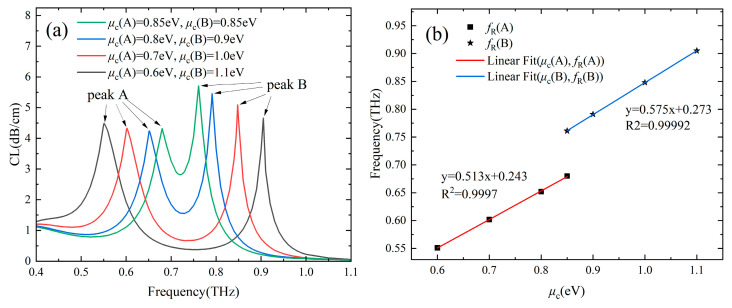
(**a**) Loss spectra with different graphene chemical potentials *μ*_c_. (**b**) Fitting line of the resonance frequency *f*_R_ as a function of *µ*_c_. (Channel A: *T* = 30 °C, Channel B: *n_a_* = 1.33).

**Figure 4 sensors-24-06225-f004:**
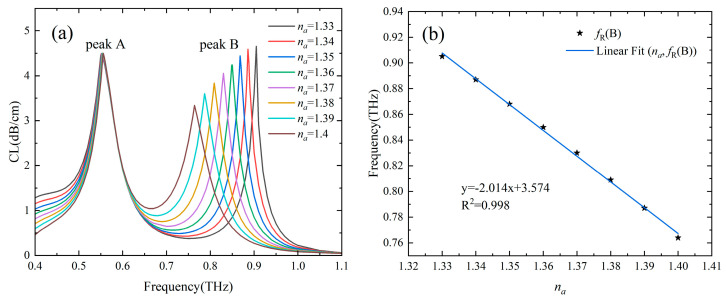
(**a**) Loss spectra of the core mode with *n_a_* in the range of 1.33–1.4 and *T* = 30 °C. (**b**) Fitting line of the resonance frequency of channel B as a function of *n_a_*.

**Figure 5 sensors-24-06225-f005:**
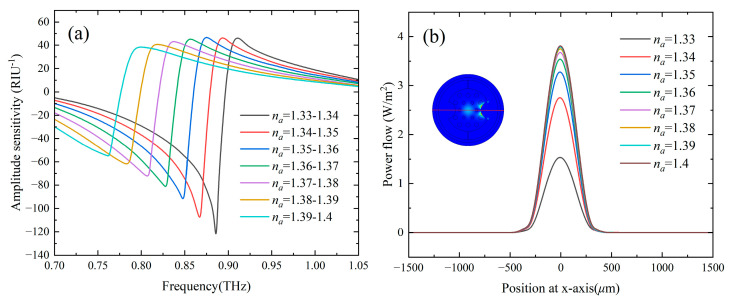
(**a**) Amplitude sensitivity of channel B, (**b**) Power flow along the x-axis for the core mode, the inset shows mode profile at the resonance frequency for *n_a_* = 1.33.

**Figure 6 sensors-24-06225-f006:**
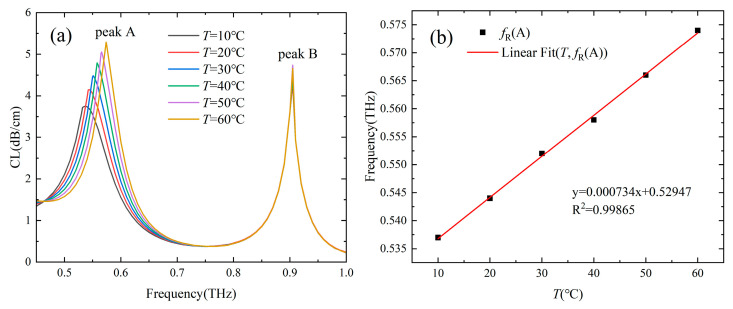
(**a**) Loss spectra of the core mode with *T* in the range of 10–60 °C and *n_a_* = 1.33. (**b**) Fitting line of the resonance frequency of channel A as a function of *T*.

**Figure 7 sensors-24-06225-f007:**
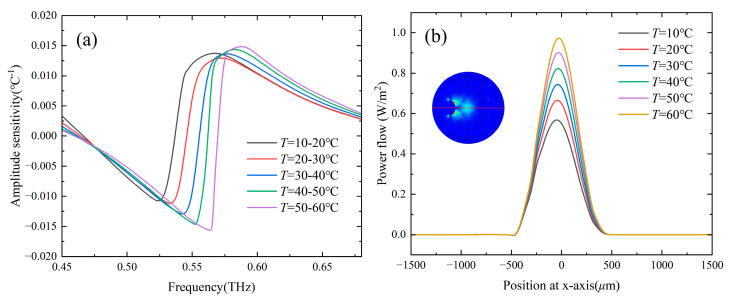
(**a**) Amplitude sensitivity of channel A. (**b**) Power flow along the x-axis for the core mode. The inset shows the mode profile at the resonance frequency for *T* = 10 °C.

**Figure 8 sensors-24-06225-f008:**
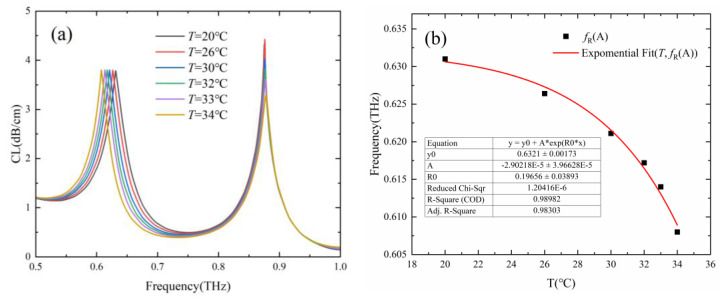
(**a**) Loss spectra of the core mode with *T* in the range of 20–34 °C and nematic liquid crystal 5CB adopted as the temperature-sensitive material. (**b**) Fitting line of the resonance frequency of channel A as a function of *T*.

**Figure 9 sensors-24-06225-f009:**
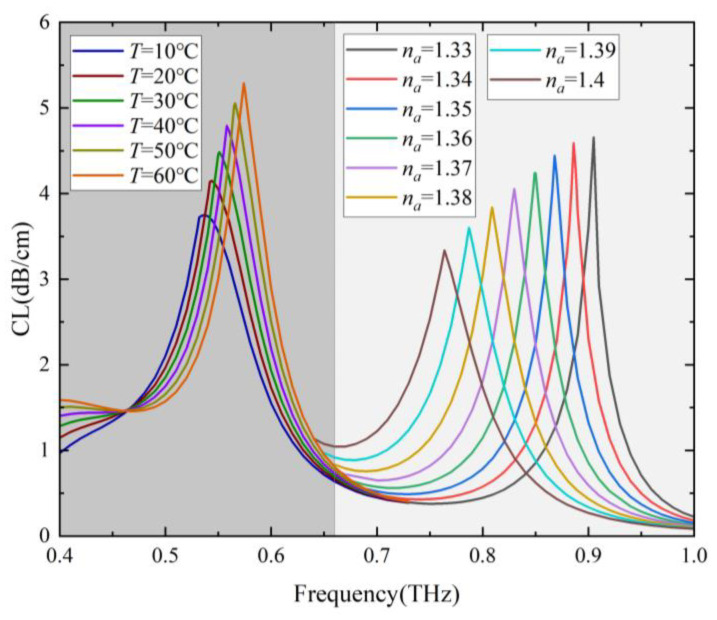
Loss spectra of the core mode with different *T* and *n_a_*.

**Figure 10 sensors-24-06225-f010:**
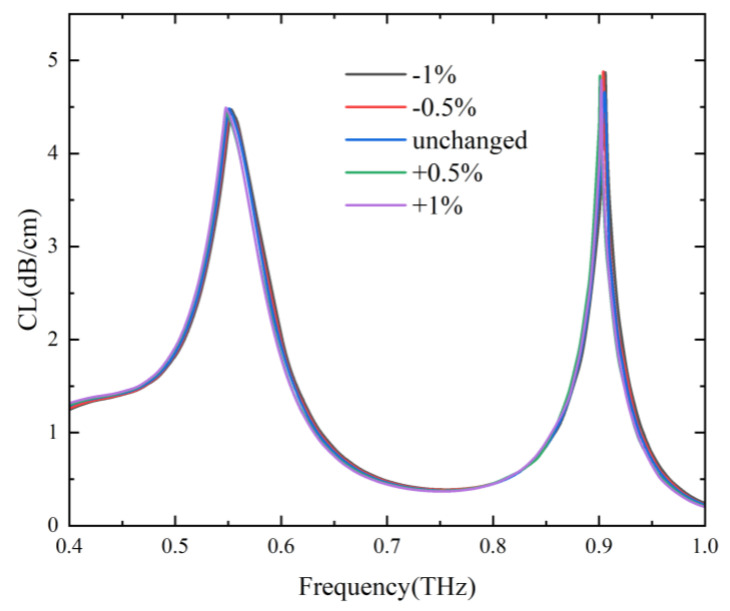
Loss spectra of the core mode with the cross-section of the designed Gr-PCF scaled by ±1%, ±0.5%, and standard size.

**Figure 11 sensors-24-06225-f011:**
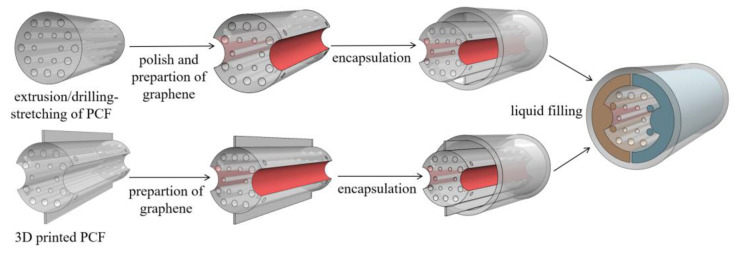
The schematic diagram of the fiber fabrication process.

**Table 1 sensors-24-06225-t001:** RI sensing performance comparison with the reported terahertz PCF-based SPR sensors.

Ref.	Sensor Structure/Plasmonic Material	Working Band	RI Range	Spectral Sensitivity	FOM (Max)
[11]	Porous fiber/PVDF	280–320 μm	1.0–1.01	400 μm/RIU	–
[12]	D-shaped PCF/MoS_2_	1.0–2.0 THz	1.33–1.4	715.59 μm/RIU	–
[13]	D-shaped PCF/PVDF	240–280 μm	1.32–1.45	335.0 μm/RIU	39.42
[15]	Dual-channel porous fiber/PVDF	300–400 μm	1.0–1.03	110 μm/RIU	–
[16]	PCF/silver nanopillars	1.1–1.19 THz	1.33–1.38	230.16 μm/RIU 1.749 THz/RIU	29.15 (FWHM = 0.06 THz)
[17]	PCF/graphene	1.1–1.5 THz	1.0–1.5	208.14 GHz/RIU	5.82
[18]	D-shaped PCF/graphene	0.45–0.75 THz	1.0–1.4	220 GHz/RIU	6.79
This work	Dual-channel PCF/graphene	0.4–1.0 THz	1.33–1.4	873.98 μm/RIU 2.014 THz/RIU	80.2

**Table 2 sensors-24-06225-t002:** Sensing performance comparison of the proposed sensor with the reported terahertz dual-parameter sensors.

Ref.	Working Band (THz)	RI Sensitivity (THz/RIU)	FOM	Q	Temperature Sensitivity (GHz/°C)
[35]	0.3–1.1	0.14	5.01	–	0.007
[36]	0.4–1.6	0.276	0.067	–	3.42
[37]	0.5–3.0	0.287	2.8	16.3	22.0
[38]	1.1–1.6	1.3	24.76	27	6.4
This work	0.4–1.0	2.014	80.2	36.2	0.734 6.0 (maximum)

## Data Availability

Data are contained within the article.

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
