# Peer review of "Terahertz Refractive Index and Temperature Dual-Parameter Sensor Based on Surface Plasmon Resonance in Two-Channel Photonic Crystal Fiber"

_sensors, 2024, doi:10.3390/s24196225_

Round 1
Reviewer 1 Report
Comments and Suggestions for Authors
The authors proposed a tunable THz sensing method based on photonic crystal fiber with two sensing channels for measuring of RI and temperature of a material. Graphene coated on the micro-grooves in the cladding was used as plasma material and introducing tunability. The dispersion relation, mode coupling and sensing characteristics of the fiber were investigated theoretically by using the finite element method (FEM).
The simulation showed a high sensitivity of 2.014 THz/RIU and 0.734 GHz/°C for analytes with refractive index 1.33 to 1.4 and environment temperature 10–60°C. The device was capable of working in the low frequency range of terahertz. It may have potential application in sensing liquid biological analytes.
Comments on the Quality of English LanguageThe manuscript was well prepared. Minor grammar corrections are need for publication.
Author Response
Dear Dr:
Thank you very much for your valuable comments. We studied your comments very carefully. Your comments are very helpful to improve our work. According to your comments and recommendations, the revisions are made below.
The original comments:
The authors proposed a tunable THz sensing method based on photonic crystal fiber with two sensing channels for measuring of RI and temperature of a material. Graphene coated on the micro-grooves in the cladding was used as plasma material and introducing tunability. The dispersion relation, mode coupling and sensing characteristics of the fiber were investigated theoretically by using the finite element method (FEM).
The simulation showed a high sensitivity of 2.014 THz/RIU and 0.734 GHz/°C for analytes with refractive index 1.33 to 1.4 and environment temperature 10–60°C. The device was capable of working in the low frequency range of terahertz. It may have potential application in sensing liquid biological analytes.
Comments on the Quality of English Language.
The manuscript was well prepared. Minor grammar corrections are need for publication.
Answer:
We have done our best to improve the language and check grammatical and typo errors. Some grammatical and typo errors have been corrected in the revised manuscript.

Reviewer 2 Report
Comments and Suggestions for Authors
This work proposes a two-channel terahertz photonic crystal fiber sensor designed for simultaneous refractive index and temperature sensing. The tunability in the terahertz range has been demonstrated numerically by incorporating a graphene coating as the plasmonic material. Since this paper only presents numerical results, it is essential to provide a more detailed presentation and analysis of the findings. Additionally, some sections require further explanation and elaboration.
(1) The simulation did not take into account any noise, so please explain how you estimated the limit of detection (LOD) for the refractive index resolution of 10^-5 as mentioned in lines 182-184.
(2) Can the temperature sensing results be correlated with the refractive index sensing results? Is there any self-correction mechanism or established relationship that can ensure the sensor performs accurately in real-world conditions?
(3) How does the sensor's performance sensitivity change with variations in hole size or misalignment of their positions?
(4) The author compares two types of channel designs with different fillings. Could other filling arrangements potentially yield better results? Additionally, in lines 137-139, the author mentions that the filling type 2 structure effectively enhances the coupling effect. Could the author provide further details and a more thorough explanation of this improvement? The differences in the mode profile images are not clearly apparent.
(5) In lines 250 and 251, the author notes that temperature sensitivity is currently low but could be enhanced by using temperature-sensitive materials. Could the author provide a list of potential materials and calculate the corresponding thermal sensitivity as evidence? This would be particularly useful since the paper lacks experimental validation.
Author Response
Please see the attachment
Dear Dr:
Thank you very much for your valuable comments. We studied your comments very carefully. Your comments are very helpful to improve our work. According to your comments and recommendations, the revisions are made below.
The original comments:
This work proposes a two-channel terahertz photonic crystal fiber sensor designed for simultaneous refractive index and temperature sensing. The tunability in the terahertz range has been demonstrated numerically by incorporating a graphene coating as the plasmonic material. Since this paper only presents numerical results, it is essential to provide a more detailed presentation and analysis of the findings. Additionally, some sections require further explanation and elaboration.
(1) The simulation did not take into account any noise, so please explain how you estimated the limit of detection (LOD) for the refractive index resolution of 10^-5 as mentioned in lines 182-184.
Answer:
The refractive index resolution of sensor is the smallest change of refractive index that can be detected. Actually, the resolution is related to the precision of the measurement, the sensitivity of the sensor and the minimum spectral resolution of the detector. The theoretical value of resolution can be calculated by the following equation:
The RI sensitivity of the designed sensor is 2.014 THz/RIU. Assuming a typical 100 MHz spectral resolution of a THz TDS setup, which is achievable for the commercial THz TDS systems, the RI resolution R = Δfmin / S = 100 MHz / 2.014 THz ≈ 4.97×10-5 RIU was obtained. So RI resolution of the order of 10-5 can be estimated.
Following sentences and Eq. (7) are added in the revised manuscript. (at the end of the second part. in lines 139-145.)
Resolution is an important parameter of sensors [29].The RI resolution of a sensing system based on frequency spectrum measurement can be expressed as:
(7)
where Δna denotes the change in analyte RI, ΔfR is the corresponding shift of the resonance frequency caused by the change of analyte RI, Δfmin denotes the minimum spectral resolution of the detector. The RI resolution can also be expressed as R = Δfmin / S by substituting Eq. (5) into Eq. (7), which depends on the sensitivity of the sensor and the frequency spectral resolution of the THz-TDS setup.
(2) Can the temperature sensing results be correlated with the refractive index sensing results? Is there any self-correction mechanism or established relationship that can ensure the sensor performs accurately in real-world conditions?
Answer:
Firstly, the designed sensor structure is relatively simple, including base material Topas, plasma material graphene, temperature sensitive liquid and analyte. In order to ensure that the sensor performs accurately in real-world conditions, the thermal optical effect of Topas (dn/dT=1 × 10-4 RIU/° C) was considered during the simulation process. Within the temperature range of 10 – 60 °C, the temperature dependence of the graphene conductivity can be ignored.
Secondly, temperature and refractive index sensing were realized using different frequency bands by adjusting the chemical potential of graphene in two sensing channels. In order to reduce the interaction between the two loss peaks and improve the detection precision, the resonance frequencies were separated efficiently by increasing the difference of graphene chemical potential between the two channels.
Therefore, the designed sensor is capable of measuring the values and changes of the environmental temperature (i.e. the temperature of the temperature-sensitive liquid as well as the analyte in the thermal equilibrium state) and the refractive index of the analyte. It can be used for detection of temperature, RI of analytes, and studying the temperature dependence of the RI in practical applications.
- How does the sensor's performance sensitivity change with variations in hole size or misalignment of their positions?
Answer:
Firstly, variation of the sensor's loss spectra with hole size has been studied during the process of optimizing the structural parameters as shown in the following Figure. In order to obtain two higher, sharper, and sufficiently separated loss peaks in the loss spectrum, which is beneficial for achieving superior sensing performance, the optimized structural parameters of d1 = 100 μm and d2 = 150 μm were adopted.
Figure Loss spectra with different hole diameter
Secondly, the fault tolerance (±0.5% and ±1% change of the hole diameter) and misalignment of the hole positions have been studied and discussed in the revised manuscript.
Following sentences and Figure 10 are added in the revised manuscript. (in section 3.4, in lines 302-313.)
In order to evaluate the fault tolerance during the preparation process of the designed two-channel Gr-PCF, the cross-section of the fiber was scaled by ±0.5% and ±1%, and the corresponding loss spectra of the core mode were shown in Figure 10. When the cross-section was changed by ±1%, the variations of the hole diameter d1 and d2 was 1 μm and 1.5 μm, respectively. It can be seen that the resonance frequencies and loss peaks change very little. What’s more, misalignment of hole position has little impact on the transmission performance owing to the refractive index guiding mechanism of the designed fiber.
Figure 10. Loss spectra of the core mode with the cross-section of the designed Gr-PCF scaled by ±1%, ±0.5% and standard size.
(4) The author compares two types of channel designs with different fillings. Could other filling arrangements potentially yield better results? Additionally, in lines 137-139, the author mentions that the filling type 2 structure effectively enhances the coupling effect. Could the author provide further details and a more thorough explanation of this improvement? The differences in the mode profile images are not clearly apparent.
Answer:
Firstly, several other filling arrangements have been designed and compared with these two types of channel designs during the optimization process. However, these two filling arrangements possess superior performances, such as better guiding property, higher FOM and relatively relatively simple filling structures.
Secondly, following sentences are added in the revised manuscript to explain the filling type 2 structure effectively enhances the coupling effect. (in section 3.1, in lines 149-160.)
It can be seen that the loss spectra of filling type 2, i.e. the two-sided selective filling structure possesses sharper loss peaks and thus superior sensing performance. By comparing the mode profiles of Figure 2(i) with Figure 2(iii) or Figure 2(ii) with Figure 2(iv), it can be seen that the filling type 2 has more core mode energy coupled into the SPP mode, i.e. higher coupling efficiency and thus sharper loss peaks. Besides, the filling type 2 has more energy located in the cladding holes filled with temperature-sensitive liquid or analytes. As the SPP modes are mainly located in the dielectrics on both sides of the plasmonic material and extremely sensitive to the changes in the RI of these dielectrics. From this point of view, the two-sided filling structure is capable of providing better sensing performance compared with the filling type 1. So type 2 was adopted in the subsequent study.
Thirdly, in order to clearly shown the differences in the mode profiles of the filling type 1 and filling type 2, Figure 2(i) and Figure 2(iii) are added in the revised manuscript and compared with Figure 2(iii) and Figure 2(iv), respectively. In order to shown the differences in the mode profiles at the resonance frequencies and other non-resonance frequencies, Figure 2(i) and Figure 2(iii) are added in the revised manuscript and compared with Figure 2(v) are added in the revised manuscript and compared with Figure 2(iii) and Figure 2(iv).
(5) In lines 250 and 251, the author notes that temperature sensitivity is currently low but could be enhanced by using temperature-sensitive materials. Could the author provide a list of potential materials and calculate the corresponding thermal sensitivity as evidence? This would be particularly useful since the paper lacks experimental validation.
Answer:
There are several materials such as strontium titanate (SrTiO3, STO), polydimethylsiloxane (PDMS), indium antimonide (InSb) and liquid crystal (LC) that can be used as the temperature-sensitive material besides the ethanol or chloroform. The frequency and temperature-dependent optical constants for the LC 5CB in THz frequency range have been studied experimentally and theoretically, providing us both reliable experimental data and theoretical models [34]. The LC 5CB has low extinction coefficient and larger temperature gradient of the ordinary refractive index dno/dT around room temperature at terahertz frequencies [30-31]. The thermal sensitivity of the designed sensor using LC 5CB as the temperature-sensitive material was calculated as evidence in the revised manuscript.
Following sentences and Figure 8 are added in the revised manuscript. (in section 3.3, in lines 259-281.)
Temperature sensitivity is mainly determined by the thermal-optical coefficient of the temperature-sensitive material. The temperature-dependent optical constants of several temperature-sensitive materials such as STO [31], PDMS [32] and liquid crystal (LC) [33] have been studied in the terahertz frequency range. The LC 5CB has low extinction coefficient and larger temperature gradient of the ordinary refractive index dno/dT around room temperature at terahertz frequencies [33-34]. The temperature sensing characteristics of the proposed Gr-PCF with 5CB as temperature sensitive material were studied and shown in Figure 8. The frequency dependence and the temperature dependence of the THz optical constants of nematic LC 5CB was based on the experimental results of pan et al [34]. The two cladding holes adjacent to the micro-grooves of channel A was filled with LC 5CB and RI of the analyte in the channel B was set as 1.33. It can be seen that the resonance frequency of channel A decreases with the increasing of temperature, while the resonance frequency of channel B remains almost unchanged. The maximum temperature sensitivity of 6.0 GHz/℃ was obtained in the temperature range of 33 °C to 34 °C. The results show that the temperature sensitivity of the designed sensor can be improved by using a filling material with higher thermal-optical coefficient. However, the mixture of ethanol and chloroform was still used as the temperature sensitive material in the subsequent study as it has advantages such as ease for filling, linear and wider temperature response range.
Figure 8. (a) Loss spectra of the core mode with T in the range of 20 °C – 34 °C and nematic liquid crystal 5CB adopted as the temperature-sensitive material. (b) Fitting line of the resonance frequency of channel A as a function of T.

Reviewer 3 Report
Comments and Suggestions for Authors
The paper “Terahertz refractive index and temperature dual-parameter sensor based on surface plasmon resonance in two-channel photonic crystal fiber” by Doudou Wang with coauthors devoted to development of sensor, which measure simultaneously refractive index and temperature. The paper is well written and present novel and interesting results. The only important comments is about insufficient argumentation of actuality of the current work. Indeed, there are many SPR sensor, which operate in the optical region [R1-R3] (shift of loss maximum happens in the spectral range near 1.55 mm). In this work author use THz radiation and it is not really clear which advantageous it provides. From the general point of view optical band is much more simple to be reach and measure rather than THz. The advantageous of utilization THz must be clearly discussed in the current paper and references [R1-R3] must be properly cited.
[R1] E. Haque, S. Mahmuda, M. A. Hossain, N. H. Hai, Y. Namihira and F. Ahmed, "Highly Sensitive Dual-Core PCF Based Plasmonic Refractive Index Sensor for Low Refractive Index Detection," in IEEE Photonics Journal, vol. 11, no. 5, pp. 1-9, Oct. 2019, Art no. 7905309, doi: 10.1109/JPHOT.2019.2931713
[R2] G. Wang, Y. Lu, L. Duan and J. Yao, "A Refractive Index Sensor Based on PCF With Ultra-Wide Detection Range," in IEEE Journal of Selected Topics in Quantum Electronics, vol. 27, no. 4, pp. 1-8, July-Aug. 2021, Art no. 5600108, doi: 10.1109/JSTQE.2020.2993866
[R3] Zhang, J.; Yuan, J.; Qu, Y.; Qiu, S.; Mei, C.; Zhou, X.; Yan, B.; Wu, Q.; Wang, K.; Sang, X.; et al. A Surface Plasmon Resonance-Based Photonic Crystal Fiber Sensor for Simultaneously Measuring the Refractive Index and Temperature. Polymers 2022, 14, 3893. https://doi.org/10.3390/polym14183893
Author Response
Dear Dr:
Thank you very much for your valuable comments. We studied your comments very carefully. Your comments are very helpful to improve our work. According to your comments and recommendations, the revisions are made below.
The original comments:
The paper “Terahertz refractive index and temperature dual-parameter sensor based on surface plasmon resonance in two-channel photonic crystal fiber” by Doudou Wang with coauthors devoted to development of sensor, which measure simultaneously refractive index and temperature. The paper is well written and present novel and interesting results. The only important comments is about insufficient argumentation of actuality of the current work. Indeed, there are many SPR sensor, which operate in the optical region [R1-R3] (shift of loss maximum happens in the spectral range near 1.55μm). In this work author use THz radiation and it is not really clear which advantageous it provides. From the general point of view optical band is much more simple to be reach and measure rather than THz. The advantageous of utilization THz must be clearly discussed in the current paper and references [R1-R3] must be properly cited.
[R1] E. Haque, S. Mahmuda, M. A. Hossain, N. H. Hai, Y. Namihira and F. Ahmed, "Highly Sensitive Dual-Core PCF Based Plasmonic Refractive Index Sensor for Low Refractive Index Detection," in IEEE Photonics Journal, vol. 11, no. 5, pp. 1-9, Oct. 2019, Art no. 7905309, doi: 10.1109/JPHOT.2019.2931713
[R2] G. Wang, Y. Lu, L. Duan and J. Yao, "A Refractive Index Sensor Based on PCF With Ultra-Wide Detection Range," in IEEE Journal of Selected Topics in Quantum Electronics, vol. 27, no. 4, pp. 1-8, July-Aug. 2021, Art no. 5600108, doi: 10.1109/JSTQE.2020.2993866
[R3] Zhang, J.; Yuan, J.; Qu, Y.; Qiu, S.; Mei, C.; Zhou, X.; Yan, B.; Wu, Q.; Wang, K.; Sang, X.; et al. A Surface Plasmon Resonance-Based Photonic Crystal Fiber Sensor for Simultaneously Measuring the Refractive Index and Temperature. Polymers 2022, 14, 3893. https://doi.org/10.3390/polym14183893
Answer:
The PCF SPR sensor working at terahertz frequency has many advantages and potential applications in bio and chemical sensing compared with the reported PCF SPR sensors working in the telecommunications and visible wavelengths.
Firstly, terahertz waves are transparent for nonmetallic and nonpolar substances compared with visible wavelengths, the most compelling feature of THz waves is the fact that a large number of atoms and molecules have rotational, translational, and vibrational responses which uniquely occur in the THz band.
Secondly, THz SPR sensors may play an important role in the field of biosensing. THz radiation allows for the direct probing of the molecular dynamics of many biomolecules as the wavelength is comparable with intramolecular vibrational modes. This made the THz SPR sensors suitable for the label-free detection of larger particles such as specific bacteria, while SPR sensors in the visible can only detect nano-molecules with sizes less than 100nm. (Ref. A. Hassani, and M. Skorobogatiy, Surface plasmon resonance-like integrated sensor at terahertz frequencies for gaseous analytes. Optics Express, 2008, 16(25):20206-14.)
Thirdly, the designed graphene based SPR sensor is dynamically tunable compared with the noble metal based SPR sensors works in the telecommunications and visible wavelengths.
Following sentences and references are added in the revised manuscript (in the introduction. in lines 36-41).
However, most of the reported PCF-based SPR sensors work in the communication or mid-IR band due to the commonly used plasmonic materials, i.e. noble metals or metal oxides possess superior performance in these frequencies [7-10]. Compared with the fiber-based SPR sensor working the optical band, its counterpart in the THz range possess many superiorities such as robust structure, ease of operation, greater penetration depth of the evanescent field, sensitive to the larger particles such as bacteria, capable of penetrating non-transparent biological samples and improving the sensitivity of biomolecules [11,12]. So THz sensors have more advantages in the field of biosensing [13].
[8] Haque, E.; Mahmuda, S.; Hossain, M.; Hai, N.; Namihira, Y.; Ahmed F. Highly sensitive dual-core PCF based plasmonic refractive index sensor for low refractive index detection. IEEE Photonics J. 2019, 11, 1-9. [CrossRef]
[9] Wang, G.; Lu, Y.; Duan, L.; Yao, J.; A refractive index sensor based on PCF with ultra-wide detection range. IEEE J SEL TOP QUANT 2020, 27: 1-8. [CrossRef]
[10] Zhang, J.; Yuan, J.; Qu, Y.; Qiu, S.; Mei, C.; Zhou, X.; Yan, B.; Wu, Q.; Wang, K.; Sang, X.; Yu, C. A surface plasmon resonance-based photonic crystal fiber sensor for simultaneously measuring the refractive index and temperature. Polymers 2022, 14, 3893. [CrossRef]

Round 2
Reviewer 2 Report
Comments and Suggestions for Authors
The revised article has answered my questions
Author Response
Answer: Firstly, We describe the abbreviations mentioned in the article in detail.
Secondly, following sentences are added in the revised manuscript to explain the filling type 2 structure effectively enhances the coupling effect. (in section 3.1, in lines 149-182.)
Figure 2a and Figure 2b show the mode dispersion curves and loss spectra of the two-channel Gr-PCF with filling type 1 and filling type 2, respectively. When the effective refractive index Re (neff) of SPP mode is equal to that of the core mode, i.e. their dispersion curves intersect, resonance coupling occurs between them, and the corresponding frequency is defined as the resonance frequency fR. At the resonance frequency, energy of the core mode couples into the SPP mode and decays rapidly, which results in a significant decrease of energy in the core region. The position of the loss peaks or the resonance frequency changes with the RI of the filling material. Therefore, RI sensing can be achieved by monitoring the position and movement of the loss peaks. From this point of view, sharper loss peaks (weaker core energy) mean better sensing performance i.e. high FOM value of the sensor. It can be seen that the loss spectra of filling type 2, i.e. the two-sided selective filling structure possesses sharper loss peaks and thus superior sensing performance. By comparing the mode profiles of Figure 2(i) with Figure 2(iii) or Figure 2(ii) with Figure 2(iv), it can be seen that the filling type 2 has more core mode energy coupled into the SPP mode, i.e. higher coupling efficiency and thus sharper loss peaks. Because the filling type 2 can achieve complete coupling between the core mode and SPP mode. Besides, the filling type 2 has more energy located in the cladding holes filled with temperature-sensitive liquid or analytes. The SPP modes are mainly located in the dielectrics on both sides of the plasmonic material and extremely sensitive to the changes in the RI of these dielectrics. From this point of view, the two-sided filling structure is capable of providing better sensing performance compared with the filling type 1. So type 2 was adopted in the subsequent study. The peak A and peak B in the loss spectra are caused by the coupling of the fundamental core mode to the SPP modes in channel A and channel B respectively, as indicated by the mode profiles in Figure 2(iii) and Figure 2(iv). It can be seen from Figure 2(v) that energy of the core mode is well confined in the core region at the non-resonance frequency. The position and intensity of the loss peak A at the lower frequency (0.552 THz) is sensitive to the RI as well as T of the temperature-sensitive liquid in the channel A, while loss peak B at the higher frequency (0.905 THz) is sensitive to the RI of the analyte in the channel B. Dual parametric sensing of temperature and RI can be realized by detecting the position and intensity of the two peaks in the loss spectrum.
